# Tuning up Transcription Factors for Therapy

**DOI:** 10.3390/molecules25081902

**Published:** 2020-04-20

**Authors:** Attila Becskei

**Affiliations:** Biozentrum, University of Basel, Klingelbergstrasse 50/70, 4056 Basel, Switzerland; attila.becskei@unibas.ch; Tel.: +41-61-207-22-22

**Keywords:** Lac repressor, Tet Repressor, TAL-Effector, transcription activator-like effector, dead Cas9, homodimerization, aggregation, dissociation rate constant

## Abstract

The recent developments in the delivery and design of transcription factors put their therapeutic applications within reach, exemplified by cell replacement, cancer differentiation and T-cell based cancer therapies. The success of such applications depends on the efficacy and precision in the action of transcription factors. The biophysical and genetic characterization of the paradigmatic prokaryotic repressors, LacI and TetR and the designer transcription factors, transcription activator-like effector (TALE) and CRISPR-dCas9 revealed common principles behind their efficacy, which can aid the optimization of transcriptional activators and repressors. Further studies will be required to analyze the linkage between dissociation constants and enzymatic activity, the role of phase separation and squelching in activation and repression and the long-range interaction of transcription factors with epigenetic regulators in the context of the chromosomes. Understanding these mechanisms will help to tailor natural and synthetic transcription factors to the needs of specific applications.

## 1. Introduction

Transcription factors (TF) determine what combination of genes a cell expresses in a given condition, at a given point of space and time. Thus, they are highly appropriate to control cellular phenotypes. Indeed, TFs have long been known to be able to reprogram one cell type into another [1]. With appropriate combination of TFs, it became possible to reprogram differentiated cell types even into embryonic stem cells [2], from which nearly any cell type can be obtained, opening the way to cell replacement therapies. This success refocused the attention to TFs as possible tools in medical therapy.

Two further discoveries increased the practical applicability of TFs; both of them revolve around the combinatorial principle. First, eukaryotic TFs turned out to be combinations of DNA binding and regulatory domains, which facilitated the design of various regulators targeted to a specific chromosomal location. Second, the target recognition of some natural and synthetic TFs is modular—each element or domain in a TF uniquely defines the nucleotide in the DNA it recognizes. Thus, a TF can be designed to recognize an arbitrary DNA sequence, simply by combining these elements. This combinatorial principle is characteristic of the transcription activator-like effectors (TALEs), the catalytically dead derivatives of CRISPR-Cas and the zinc-finger TFs [3].

With TFs being able to target arbitrary sequences, it has become possible to optimize the binding affinity and specificity using kinetic-biophysical principles, analogously to the recent efforts to optimize the affinity or the rate of the binding of small molecule drugs to their receptors employing biophysical principles, which improved the drug discovery and design [4,5].

Since many reviews have focused on the biotechnological optimization of TFs, this review focuses on how their genetic-biochemical and biophysical properties affect the efficacy and specificity of designer and commonly used prototypical TFs. Following the brief introduction of these TFs, their possible applications are listed, in order to appreciate the needs of the TF optimization.

## 2. The Modularity of Transcriptional Regulators

### 2.1. The Modularity Principle in Sequence Recognition

#### 2.1.1. The TAL Effector

Pathogenic bacteria resemble heavy force combatants, equipped with a multitude of weapons and defense shields. On the other hand, viruses possess only few genes, which makes them depend on the host cell, in that they act as sophisticated agents that reprogram the cell. The plant pathogenic bacteria from the genus Xanthomonas imitate in this sense the viruses, since they use an elaborate scheme to reprogram the host plant cell. They possess transcription activator-like effectors (TAL effectors, TALEs), proteins with the ability to directly bind the promoters of genes in the host. The control of the expression of the host genes helps bacterial colonization, inasmuch as TALEs trick the plant into activating weak points that allow an invasion. This has been possible thanks to the remarkable DNA binding mechanism, known as the TALE code—each base pair is recognized by a specific repeat (protein domain) in the TALE [6,7,8]. Amino acids at positions 12 and 13, termed repeat variable di-residues (RVDs), determine the base preferences of a repeat.

#### 2.1.2. The CRISPR-Cas System

The second family of DNA-binding proteins with a remarkable modularity in the sequence recognition comprises the Cas endonucleases, which are also of bacterial origin. The genomes of most Bacteria and Archaea harbor Clustered regularly interspaced short palindromic repeats (CRISPR), which are involved in resistance to bacteriophages. When bacteria encounter bacteriophages, they integrate sequences derived from phage genomic sequences. Removal or addition of such sequences modifies the phage-resistance phenotype of the cell [9]. These DNA sequences are transcribed into the CRISPR RNA, which binds to the Cas9 protein. This heterodimer binds to the target DNA, which is complementary to the CRISPR RNA and cleaves the DNA preventing new infection by the phages [10]. CRISPR-Cas9 from *Streptococcus pyogenes* is possibly the most well-characterized CRISPR-Cas system, which has been harnessed for genome editing in many eukaryotes. Cas9 has been also repurposed for transcriptional regulation, relying on the catalytically inactive Cas9 variant, dead Cas9 (dCas9) [11].

#### 2.1.3. Zinc Finger Proteins

The above examples may convey the idea that prokaryotes are the major source of TFs with a modular design. However, the third major class of modularly designed TFs relies on zinc fingers, which are widespread in eukaryotes, even though more recently bacterial zinc fingers have been also identified [12]. Typically, two cysteine and two histidine residues coordinate a zinc ion to form a compact structure that determines the DNA sequence to be recognized. Although the creation of zinc fingers was a critical advance in gene editing and the design of synthetic TFs, their use has grown less rapidly because of the engineering challenges associated with context- dependent assembly constraints [3].

#### 2.1.4. Non-Modular Prototypic DNA Binding Proteins—The Lac Repressor (LacI) and the Tet Repressor (TetR)

Two further TFs are included in this review that do not follow the modularity principle in sequence recognition but have played a major role in understanding transcriptional regulation and were widely used in a variety of organisms—LacI and TetR.

LacI is expressed at low copy numbers and regulates the genes in response to lactose in *E. coli*. Upon complexing with lactose, LacI dissociates from the operator sequence in the *lacZYA* promoter, which leads to the expression of proteins involved in lactose uptake and metabolism [13]. Besides being one of the first TFs to be discovered, it can be conveniently controlled with lactose or lactose analogues, which contributed to its widespread use.

The TetR repressor regulates the expression of the TetA tetracycline pump, a key determinant of bacterial resistance against tetracycline antibiotics. When a tetracycline permeates the cell membrane, it binds to the TetR, which then dissociates from the *tet* operator in the promoter of the *tetA* gene, enabling a high expression of the pump, which then pumps out the antibiotic [14]. A few mutations in the TetR amino acid sequence were sufficient to switch the behavior of TetR with respect to tetracycline binding; unlike the TetR, the reverse TetR mutant associates with the *tet* operator upon being complexed with tetracycline [15].

### 2.2. The Modularity Principle in Eukaryotic Gene Expression: The Convenient Conversion of a Prokaryotic Repressor into a Eukaryotic Activator

Eukaryotic transcriptional activators are modular consisting of a DNA binding domain and transcriptional activation domain, which stands in contrast to prokaryotes. Prokaryotic transcriptional activators, with few exceptions, are not modular [16].

The modularity was unveiled in the model eukaryotic organism budding yeast, by examining various chimeras of the potent transcriptional activator Gal4. When the DNA binding domain of the Gal4 was replaced by the *E. coli* LexA repressor, the resulting hybrid LexA-Gal4 fragment was fully capable of activating transcription in yeast [17]. In this case, the prokaryotic repressor was acting solely as a DNA binding domain that tethered the activation domain to the DNA. This convertibility of prokaryotic repressors into functional DNA binding domains in eukaryotes relies on the fact that most prokaryotic repressors simply act as a roadblock in the regulation of prokaryotic gene expression, blocking the passage of the RNA polymerase (Figure 1). This logic is different from eukaryotic repressors which can interact with the polymerase even when they are bound upstream of the transcription initiation complex and do not block the passage of the polymerase [18].

A study aimed at the systematic identification of activation domains revealed that modularity is widespread but there are exceptions to this rule; activation domains were found that overlap with structured DNA binding domains [19]. The activation domains of different TFs are typically enriched in a specific type of amino acids, such as histidine, proline, acidic amino acids or glutamine.

The modularity of transcriptional factors permits a wide range of combinations by fusing a DNA binding domain with regulatory domains, including activation and repression domains, such as the VP64 activation and KRAB repression domains [20]. Ongoing search for high activation potential across multiple cell lines has led to the design of novel activation domains. The VPR and SunTag have higher activation potency than VP64 when fused to TALEs or dCas9 [21,22]. These activation domains can be modulated by fusing them to the estradiol receptor, which renders the activation inducible by the estradiol analogue 4-OH-tamoxifen [23]. For example, when the expression level of CARs was controlled with tamoxifen, it was possible to modulate the killing activity of lymphocytes [24].

DNA binding domains can be also fused to endonucleases, which enable genome editing even in mammalian genomes. In addition to the DNA, RNA can be also targeted. Recent discoveries of the new Cas family members have extended the scope of the applications, exemplified by targeted RNA degradation and RNA-based manipulations, which rely on Cas13a, an RNA-guided RNA ribonuclease [25].

## 3. Potential Therapeutic Applications of TFs

The therapeutic application of TFs depends on their delivery into the cells and on the extent to which the expression of genes can be modulated. For most genetically inherited diseases, the replacement of the defective gene or the introduction of a healthy gene into cells that compensate the loss of gene function, provides the most direct solution. The first human gene therapy relied on such curative gene expression—a variant of the lipoprotein lipase was expressed in muscle cells to compensate the defective gene (Table 1) [26].

A more direct application of TFs to correct gene expression is appropriate in cases when a function declines in a pathophysiological condition without having mutations in the underlying genes. Such a goal was set when a zinc-finger TF was targeted to the VEGF promoter, in order to express the VEGF. The increased VEGF expression leads to the formation of new blood vessels and the regeneration of the microvasculature, which is compromised in diabetic neuropathy (Table 1) [27].

TFs can be used in theranostic devices, which can diagnose a disease state and trigger an autonomously regulated therapeutic response. For example, a cell expressing a bacterial uric acid sensor is a theranostic device relevant for the treatment of diseases associated with hyperuricemia, like gout. In these cells, the uric acid concentration is converted by a TF to the appropriate production rate of urate oxidase, which then can control uric acid concentration in hyperuricemic mice [28].

TFs are expected to be applied to cell-based cancer therapies, as well. Patient T cells are harvested and engineered to express a cancer specific chimeric antigen receptor (CAR) (Table 1). In most T cell therapies, a single cancer antigen is targeted but it would be advantageous to detect specific combinations of antigens and other markers. In this case, the TFs can perform specific logical operations that decode multiple inputs and generate a single output to activate a T-cell response to kill a cell [29]. The reengineering of T cells may be even more thorough—to overcome the effects of an immunosuppressive microenvironment, a frequent condition in cancers, the T cells can be modified to additionally express immune-modulatory proteins, including ligands and cytokines under the control of synthetic factors [30].

New developments in the delivery methods (see below) may enable the application of TFs to differentiate cancer cells or to counteract the progression of the dedifferentiated cancer cells [36]. The expression of appropriate TFs can induce cell differentiation in various cancer models (Table 2) [37,38]. Targeting a TALE or CRISPR-dCas9 based transcriptional activator to the promoter of such a TF (e.g., Ascl, HNF-4α) could induce, in principle, the differentiation of cancers in vivo. It addition to genetic interventions, small molecules can also induce TFs to differentiate cancer cells (Table 2).

## 4. Delivery and Construction of TFs and Endonucleases

The majority of pharmacological therapies rely on small molecules, ranging from antimicrobial antibiotics to antipsychotic drugs. With the progress in molecular biology, macromolecules, such as antibodies and peptide or protein hormones, have found their way into drug therapy. Since their primary point of action is in the extracellular space, they can be injected, which simplifies their delivery. On the other hand, TFs, as well as endonucleases targeting the genome, are exerting their effect intracellularly, which poses a major challenge to their delivery. Nonetheless, recent technological advances have opened up new avenues for their delivery.

There are three main possibilities to introduce a TF into a cell—as a protein or in the form of a DNA or RNA encoding the TF.

The DNA or RNA can be introduced into the cells ex vivo for the cell-based therapies (Table 1), with standard laboratory techniques, such as transduction or transfection, exemplified by the injection of the mRNA encoding a TALEN into the isolated cells or embryos [35,41]. The chimeric antigen receptor is typically introduced by lentiviral vectors [42]. Patients that do not have sufficient healthy T cells require donor T-cells with appropriate gene deletions to prevent host response. A TALEN has been used to perform such a deletion in a clinically approved T-cell therapy. If the introduced TALEN is encoded by DNA and not by RNA, its expression must be tightly regulated because off-target effects, especially in the case of the CRISPR-Cas9 system, may result in undesired cuts. The expression of the endonuclease can be controlled by another TF, for example, the TetR based rtTA [43].

The introduction of TFs by injection precludes long-term effects. For example, the TALEN protein injected by the bacterial type III secretion system is degraded and disappears 12 h after the bacterial injection into human cells [44]. Cas9 is a stable protein. The reduction of the level by protein degradation may be advantageous in this case because it can attenuate undesired genome editing [45]. The TFs can be also tethered to protein transduction domains, which can cross the cell membranes [46].

In addition to the delivery, the construction of the designer TFs itself has been optimized because the speed of construction is a major determinant of the popularity of a technology. A major drawback of the TALE technology over the CRISPR is the relatively long time it takes to construct the long and repetitive DNA sequence. The ligation alone requires around 5 days. On the other hand, TALEs have superior regulatory features as they more easily act both as activators and repressors in comparison to CRISPR-Cas9 (see below). Furthermore, the clinical application of TALEs underscore their biotechnological relevance. The first therapeutically applied chimeric antigen receptor (CAR) T cells were engineered with TALENs in combination with lentiviral transduction [35]. A recent study has reported that the assembly of the TALE sequence can be reduced to one day [21], which may facilitate the use of TALEs. The limitation of this approach is that it is streamlined for TALEs recognizing an 18 bp DNA sequence but this is adequate for most applications because this length ensures optimal binding specificity (Figure 2).

## 5. Biotechnological Applications

In addition to medical therapy, the designer DNA-binding proteins are of major interest in biotechnological applications. The DNA binding domains can be used, for example, to recruit enzymes to the DNA in order to increase the local concentration of enzymes. In this case, the DNA serves solely as a scaffold. When TALEs tether multiple enzymes that belong to one pathway to the DNA, the production rate of the final metabolite can be enhanced [48].

The TALEs fused to GFP can be used as a DNA staining agent to monitor enzymatic reactions. With a TALE-GFP fusion that recognizes a 7-bp DNA, the endonucleotic cleavage of single DNA molecules was monitored in real time in physiologically relevant conditions [49]. In this study, a non-specific binding to AT-rich sequences was observed in addition to the above 7-bp, which is likely to reflect the observation that TALE binding to the DNA is nonspecific in the absence of magnesium [50]. The advantage of TALE-fluorescent protein fusions over classical DNA stains is that most intercalating DNA dyes generate radical oxygen species or DNA strand breaks upon irradiation during their detection and can affect the interpretation of real-time observations of cleavage reactions.

## 6. Biophysical and Molecular-Genetic Properties of the Activator and Repression Domains

The modular nature of the eukaryotic activation and repression domains facilitated the identification of their molecular genetic interactions and their biophysical analysis. These studies revealed two phenomena characteristic of eukaryotic TFs. First, they influence transcription both directly and indirectly via the epigenetic modification of chromatin. Second, they are enriched in intrinsically disordered protein regions.

The prokaryotic DNA is naked while the eukaryotic DNA is wrapped around histones. In line with this difference, the mass ratio of the basic proteins to DNA is 50 times higher in eukaryotes than in prokaryotes [51]. The naked prokaryotic DNA is easily accessible to the transcriptional machinery while the default state of the eukaryotic DNA is restrictive due to lower accessibility of the DNA in the chromatin [52]. Therefore, transcriptional activators are essential to recruit the RNA polymerase and to initiate transcription in eukaryotes, while they fine tune the gene expression in prokaryotes, playing a subordinate role in relation to repressors. The eukaryotic TFs recruit enzymes that modify chromatin, including acetylation and methylation and these epigenetic modifications influence the DNA accessibility. Thus, indirectly, repressors usually render the DNA less accessible while activators have the opposite effect.

Eukaryotic TFs have the remarkable feature of having a high content of intrinsically disordered protein regions, which shows interesting parallels with the modularity. The degree of disorder is significantly higher in eukaryotic TFs than in their prokaryotic counterparts. Secondly, the degree of disorder in activation domains is much higher than that in the DNA-binding domains [53]. Recent biophysical studies have revealed that these activation domains tend to form condensates, akin to equilibrium phase separation [54], which is supported by two types of evidence. First, these structures are disrupted by 1,6-hexanediol, which impairs hydrophobic interactions. Second, fluorescence recovery after photobleaching (FRAP) revealed that molecules move in and out of these condensates rapidly, indicating that the components that make up these structures are dynamic and not solid aggregates.

Interestingly, repressors that contain poly-glutamine (Poly-Q) repeats also have the propensity to form aggregates. The function of poly-Q-containing Ssn6 increases with its repeat number until a certain threshold where further expansion leads to aggregation [55]. In this case, the Ssn6 repressor can propagate as a prion [56]. Interestingly, the RNAs encoding proteins with polyQ-repeats also promote phase separation and form RNA droplets [57].

The intrinsic disorder makes the proteins more prone for aggregation. However, the structural disorder does not have only negative, pathologic connotations because these regions make the TFs druggable [58]. Furthermore, these regions can bind to a large number of protein interaction partners, providing a larger regulatory flexibility. This flexibility in the interactions and the interaction of the TFs with epigenetic regulators enable eukaryotic TFs to act at distance. Such an action at distance permits eukaryotic repressors to inhibit transcription even from upstream of the transcriptional initiation sites whereas their prokaryotic counterparts can act as a roadblock at or downstream of transcriptional initiation (Figure 1).

The expression of transcriptional activators, especially those with potent transcriptional activation domains, can be toxic to the cells, due to squelching. Squelching leads to an inhibition of gene expression by an activator. It is unclear whether squelching is related to aggregation or other forms of phase separation. Since squelching acts through the sequestration of mediators of transcription [59], it is possible that, in principle, the sequestrated molecules form aggregates. It has been observed that transcriptional activators including the potent VP16 activation domain often drive biphasic expression—they induce transcription shortly after their induction, followed by a decline [60,61]. Thus, squelching seems to have delayed onset, which is reminiscent of the delayed onset of aggregation, since aggregates grow in size as a function of concentration after a long nucleating lag phase [62].

## 7. Biophysical and Binding Properties of TFs

Before detailing the binding properties of the individual TFs, this section starts with the kinetic principles to remind the reader that binding constants depend on the equations used for the fitting.

Assuming that the TF is a dimer, the rate constants of the homodimerization of the monomeric protein M and the dissociation of the TF, denoted by $h_{a}$ and $h_{d}$, respectively, determine the available TF concentration (Equation (1)). The dimeric TF (TF) binds to the DNA to yield the heteromolecular complex (C). The association (binding) and dissociation (unbinding) rate constants of this binding reaction are denoted by $k_{a}$ and $k_{d}$, respectively (Equation (2)):(1)$$\frac{d[TF]}{dt}=h_{a}{\left[M\right]}^{2}-h_{d}[TF]$$
(2)$$\begin{array}{l} \frac{d[C]}{dt}=k_{a}[DNA][TF]-k_{d}[C], \end{array}$$

The binding affinity is typically expressed in terms of the equilibrium association constant or its inverse, the equilibrium dissociation constant, having molality units (Equation (3)):(3)$$K_{D}=1/K_{A}=k_{d}/k_{a}$$

The lower the value of $K_{D}$ the stronger the binding. Micromolar binding is considered weak, nanomolar is intermediate, whereas picomolar is strong. The residence time is the inverse of the dissociation rate constant, (Equation (4)), having time units, typically seconds or minutes.
(4)$$\tau =1/k_{d}$$

Most TFs are not monomeric but form dimers or even tetramers. For a dimeric TF, the dimeric form (TF) will represent only a fraction of the total protein amount (Equation (5)):(5)$$[TF]=h_{a}{\left[M\right]}^{2}/h_{d}={\left[M\right]}^{2}/K_{D,Dim}.$$

The value of the estimated binding affinity of a dimeric TF to the DNA depends on whether the dimerization reaction has been taken into account or not because the proportion of the dimer depends on the total protein concentration. Since not all studies measure or estimate the dimeric form explicitly, the equilibrium binding constants and the association rate constants, may not be directly comparable.

Weak dimerization does not necessarily entail a weak transcriptional activation or repression. For example, the yeast Gal4 is possibly the most potent activator in yeast when it binds to multiple sites in a promoter, yet its dimerization constant (*K_D_* = 20 µM in vitro and *K_D_* = 8.5 µM in vivo) reflects a rather weak binding [63,64].

Dimerization of a TF can amplify the signal transmitted to gene expression when the protein concentration of the TF is relatively low. In this case, a small change in the TF concentration generates a large change in the output. This signal amplification due to dimerization (multimerization) can be used to generate a switch-like response to a sugar and can promote cellular memory [65]. The heterodimerization of the CRISPR guide RNA with the Cas9 also has the potential for signal amplification and nonlinear reaction response, which may explain why synthetic CRISPR based gene regulatory networks display robust memory and oscillations [66].

### 7.1. TALE Activators and Repressors

TALEs bind double-stranded DNA, In addition, they can bind DNA:RNA hybrids, with a slightly lower affinity [67]. The binding affinity to the DNA can vary broadly. In a systematic study, designed TALEs were targeted to 18 bp long DNA sequences and some of the TALEs displayed a high binding affinity for their target sequence (*K_D_* = 0.16 nM) [68]. Kinetic studies indicate a relatively fast binding–unbinding reaction, with similar parameter values obtained from in vitro and in vivo measurements—the residence time ranged from 3 to 16 s (Table 3) [69,70]. When the respective TALEs were fused to an activation domain (VP64), the TALEs with the highest affinity to the target sequence yielded also the highest gene expression [68]. However, the correlation between the in vitro binding affinity and gene expression is moderate—the observed expression was strong in a range from intermediate to high affinities.

A comparison between the TALE and TetR based activation system further underscored the high affinity and the potency of the TALE to control gene expression. When *tet* operator-specific TALEs, with an identical DNA-binding site as the Tet repressor (TetR), were created, the DNA-binding domain of tetTALE alone effectively counteracted trans-activation mediated by the potent *tet* trans-activator [71].

Importantly, the highest affinity TALE is not necessarily the optimal solution for all needs since the binding to nonspecific DNA correlates with the binding affinity to the target DNA (Figure 2). The relative binding to the non-specific sites depends on the number of TALE repeats. The non-specific binding reaches a minimum when the TALE contains 18 repeats, when it is around 30 times weaker than the binding to the specific sites. For shorter and longer arrays of TALE repeats, this ratio is around 10.

The ions in the aqueous solution strongly affect the binding to nonspecific sites. TALEs demonstrate high sequence specificity only upon addition of small amounts of certain divalent cations (Mg^2+^, Ca^2+^) [50]. On the other hand, under purely monovalent salt conditions (K^+^, Na^+^), TALEs bind to target and random DNA sequences with nearly equal affinity. This effect was confirmed with TALEs having various numbers of repeats.

The reduction of the positive charge of the TALE protein also enhances the binding specificity; for example, the mutation of lysine and arginine residues to glutamine in the TALE protein decreases the nonspecific binding to the negatively charged DNA [72].

Even though the DNA binding part of TALE is an array of repeats, a polarity effect breaks the symmetry of the array—the *N*-terminal repeats recognizing the 5′ end of the target sequence contribute more to the affinity than *C*-terminal ones [68]. The TALE proteins are capable of rapid diffusion along DNA using a combination of sliding and hopping behavior [73]. The *N*-terminal region of TALEs is required for the initial non-specific binding and subsequent rapid diffusion along the DNA, whereas the central domain comprising the repeats is required to recognize the target sequence.

The binding to the target sequence can be also controlled by conformational stress. By inducing dimerization of the proteins connected to the *N*- and *C*-terminal domains of the TALE, the circularization of the protein locks a strained conformation of the protein, which leads to a reduction of transcriptional activation by TALE activators [74].

### 7.2. CRISPR-Cas

Three molecules interact to yield the Cas9-CRISPR RNA-DNA complex and the binding to the DNA is followed by an enzymatic reaction, the DNA cleavage. Thus, a more complicated kinetics is expected for this reaction than for the bimolecular binding reaction involving the TALE and DNA. The first kinetic analysis of the Cas9-CRISPR complex revealed an even more complicated kinetics—the DNA cleavage failed to obey the Michaelis-Menten kinetics [75]. The cleavage reaction stopped soon after mixing the components, leaving a large proportion of DNA uncut, even though a usual enzymatic reaction proceeds until completion. A higher proportion of DNA was cleaved only if the Cas9-RNA concentration was increased. These observations indicate that Cas9 is a single turnover enzyme that remains tightly bound to the DNA after the cleavage reaction [75].

Interestingly, so far only one enzyme has been identified among the Cas9 homologues that acts as multiple turnover enzyme, the Cas9 isolated form *Staphylococcus aureus* [76].

It is difficult to measure the kinetic parameters of single turnover enzymes. Gong et al. tackled this problem by separating the binding and the enzymatic reactions, specifically, by adding magnesium only after the completion of the binding reaction. The Cas9-gRNA complex was first incubated with radiolabeled DNA in the absence of Mg^2+.^ After formation of the Cas9-gRNA-DNA complex, an excess of unlabeled DNA was added and the DNA cleavage was initiated by adding magnesium after various times of incubation (Table 4). The dissociation rate constant was fitted to single-exponential decay function, yielding a mean residence time of 5 min [77]. This residence time is substantially longer than the few seconds measured for the TALEs (Table 3 and Table 4).

The residence times were even longer in vivo—more than 3 h in mammalian cells and between 40 and 120 min in Staphylococcus [78,79]. The in vivo measurements also indicate a relatively short half-life (15 min) of the guide RNA [78], which suggests that the guide RNA may be a limiting factor if not expressed at sufficiently high level. The length of the guide RNA is also an important determinant of the binding. When the guide RNA length is truncated from 11 to 8 nucleotides, the residence time decreases 10 times [78], in agreement with an earlier study using designer activators, which showed that guide RNAs 12 and 20 nucleotides in length generated comparably high gene expression, whereas those shorter than 8 nucleotides had negligible effect on gene expression [80].

### 7.3. The Prokaryotic Repressors LacI and TetR

The LacI and TetR are classical prokaryotic DNA binding proteins widely used in eukaryotes, which raises the question of whether their popularity can be traced back to their biophysical properties.

The capacity of LacI and TetR to act as activators or repressors in eukaryotes was compared directly in yeast. The tetR-VP16 AD activated gene expression but the LacI-VP16 failed to do so [81]. When examined whether they can act as a roadblock to repress expression driven by a transcriptional activator, LacI was able to repress transcription, albeit with a lower efficiency than TetR.

The above finding is underscored by the fact that the TetR-VP16 is more frequently mentioned in publications than the LacI-VP16 (see Methods). The analysis of the publications shows that mammalian cells and tobacco proved to be suitable hosts of the LacI-VP16 or its variants, suggesting their feasibility.

In mammalian cells, LacI was fused to a nuclear localization signal and activated gene expression when lac operators were positioned either upstream or downstream of the transcription unit [82]. Promoters containing 14 or 21 *lac* operator sequences were induced around 1,000-fold. Activation was inhibited by isopropyl-beta-D-thiogalactoside, confirming the inducibility of the protein. The fusion protein was bifunctional, also acting as a repressor when the promoter contained an operator immediately downstream of the TATA box. In a more recent study, using mammalian embryonic stem cells, Gal4-VP16 activated gene expression while LacI-VP16 failed to do so [83].

Both the DNA binding and the activation domains were modified to obtain a potent activator in tobacco. The VP16 was replaced by the Gal4 activation domain because of its higher potency [84], whereas lacIhis mutant was acting as the DNA binding domain, as the Y17H mutation is estimated to bind lac operator sequences with at least 100-fold greater affinity than the wild-type lac repressor.

The above findings indicate that LacI and TetR have similar efficiencies to act as roadblock repressors but the LacI-activation domain fusions have a lower potential to activate gene expression and require a more thorough optimization. A similar conclusion can be drawn from the in vivo and in vitro binding measurements.

*K_D_* = 0.18 nM was estimated for TetR-DNA binding with surface plasmon resonance [15] and *K_D_* = 2 nM was fitted from stop flow data based on fluorescence measurements [85]. In its active state, the reverse TetR binds the DNA with a lower affinity, *K_D_* = 10 nM [15]. The control of TetR and reverse TetR by tetracycline still represents an advantage over TALE and CRISPR because the rapid dissociation permits the measurement of rapid post-transcriptional processes, such as the degradation of RNAs with very short half-lives [86,87].

For LacI binding it is more difficult to fit the individual dissociation constants due to the linkage of the equilibria between protein monomers, dimers and tetramers [88]. In vitro binding data indicate a strong binding, with picomolar dissociation constants while in vivo data suggest a binding in the nanomolar range in *E. coli* [89]. An even higher discrepancy was found between the in vivo and in vitro data in yeast, where LacI-GFP fusions bound the DNA with micromolar dissociation rate constants [90]. This weak binding was shown to be sensitive to the nature of protein fusions [90], which possibly explains why a more careful design of the activation domain linkage is needed for LacI fusions.

## 8. Methods

In order to compare the usage of LacI and TetR based activators, we searched for publications using the keywords LacI-VP16 and TetR-VP16. “TetR-VP16” retrieves 12 and 489 hits in PubMed and Google Scholar, respectively. LacI-VP16 retrieves 0 and 27 hits, respectively. Out of the 27 hits, 4 publications use lacI-VP16 as a TF (see Section 7), while several other publications explore the role of VP16 in inducing epigenetic changes in the chromatin.

## 9. Conclusions—Optimization of Transcriptional Activation and Repression

The transcription factors TetR, LacI, CRISPR-Cas and TALE, covered in this review have been widely used in a range of organisms with varying efficiency and specificity. Their biophysical and genetic characterization allows guidelines to be formulated how to optimize TFs to activate or repress gene expression, especially with regard to the following three aspects—the intracellular concentration, the binding affinity to the DNA and the integration of the TF activity into the chromosomal regulatory landscape.

Increasing the expression of a TF is an enticing, simple option to boost transcriptional control but the outcome can quickly turn into the opposite. This is particularly true for the activation domains due to squelching (see Section 6). Furthermore, chromatin, activation- and repression-domains, can promote phase separation and the formation of aggregates, which are known to occur at and above a critical concentration [91]. Above the critical point, unintended consequences can occur. Thus, it can be advantageous to keep the intracellular concentration of the TF at intermediate or low level, for example, by reducing translation, by increasing the degradation rate of the protein or by using appropriate promoters [61,90]. An increase of the degradation rate affects the TF bound to the DNA, reducing its local effect; therefore, using a weaker promoter to drive the expression of the TF may be the preferred option [90]. At low expression, noise usually becomes prominent. Nonetheless, a variety of promoters are available with distinct stochastic properties [92].

When a TF is expressed at low level, high binding affinity to the target genes is indispensable for gene control. All main families of TFs discussed in this review bind the target DNA with high affinity, with dissociation constants in the picomolar to low nanomolar range (Section 7). This strong binding has been certainly a key factor behind their successful application in a multitude of host organisms. In pharmacology, the affinity of a drug to its target strongly correlates with the biological effect [5]. In the case of TFs, this correlation may be less strong due because a TF interacts with multiple molecules, besides the primary target, the DNA. Indeed, TALEs more efficiently activate gene expression than CRISPR-Cas9d-based activators, despite the lower physical binding to the DNA [20,93]. On the other hand, CRISPR-Cas9d based repressors are highly efficient. Similarly, TetR based activators are more efficient than LacI-based activators even though their repression efficiency is similar (Section 7.3). This larger variability in the potency of designed transcriptional activators may reflect more pronounced steric constraints in the design of the transcriptional activators or a more fundamental difference in the kinetics of activation and repression.

The kinetic parameters (binding and unbinding rate constants) drew a considerable attention in the design of targeted endonucleases. Kinetic models suggest that the cleavage of off-target sequences can be reduced by accelerating the dissociation of the enzyme from the DNA or by reducing its catalytic activity [94]. Following these predictions, an enhancement of cleavage specificity has been realized by designing enzymes with attenuated cleavage rate or by fusing Cas9 to inhibitor proteins [95,96]. In the above model, the specificity is affected by both the binding and catalytic reactions because they are coupled, since both of them involve the same molecular species, the enzyme. It is not clear whether such a coupling is present in transcription and thus similar principles apply to the specificity in transcriptional regulation. The binding of the TF to the DNA and the initiation of transcription by RNA polymerase, an enzymatic reaction, are distinct molecular reactions. However, some enzymatic reactions in transcription bear a resemblance to single turnover enzymes. There is experimental evidence that TFs, especially activators, recruit the proteasome during transcriptional activation [97]. Thus, the initiation of transcription by an activator may be followed by its proteolysis, which is reminiscent of the single turnover enzymes. CRISPR-dCas9 is thought to elicit more off-target transcription than TALE activators, which would follow from the above mechanism since the residence time of CRISPR-Cas9 is much longer (Table 3 and Table 4). However, there are studies that suggest that the two TFs affect off-target genes with similar frequency [93].

A comprehensive theory of how binding kinetics affects gene expression is yet to come but some TFs have been analyzed in detail. A study on the yeast Rap1 activator suggests that accelerating the binding-unbinding events reduces transcriptional activation, even if the equilibrium binding does not change in chromatin immunoprecipitation assays [98]. Variations in binding kinetics can also affect the timing of the gene response [99].

In addition to the classical off-target effect, caused by the binding of the TF to the off-target genes, alternative mechanisms, not involving direct binding, can play an important role, as well. Transcriptional control in higher eukaryotes is known to display long-range effects and a transcriptional activator can induce gene expression even megabases away from its binding site. This off-target activation can vary with cellular differentiation, as shown by a study on the protocadherin gene array, whose genes are expressed in neurons but not in embryonic stem cells and neuronal progenitors [100]. In stem cells, TALE activators targeted to a specific protocadherin isoform activate primarily the target gene. However, off-target genes adjacent to the target genes became activated prominently in neuronal progenitors [101]. This effect is mediated, at least in part, by epigenetic mechanisms, such as DNA demethylation, which is induced by the strong activation domain. Thus, as cells differentiate from stem cells to neurons, the epigenetic activity gradient around the recruited TALE-VP160 broadens. The VP16 (or its variants VP64, VP160) domain, which contains a large number of negatively charged amino acids, may cause these epigenetic changes independently of transcription, since the recruitment of negatively charged peptides to the DNA can cause large-scale chromatin remodeling, without inducing transcription [102].

Transcriptional control can be supplemented with epigenetic one to fortify the gene response. When the CRISPR-dCas9 activator was corecruited with the TET1 enzyme, which catalyzes the first step in the DNA demethylation, the gene activation was substantially increased [103].

With designer TFs, the 3D structure of the chromatin can be also controlled. Two studies have shown that the repression efficiency of TALEs, which act as roadblocks to repress transcription, can be enhanced when TALE fusion proteins with dimerization domains are recruited to two sites in the chromosome [104,105]. Consequently, the intervening DNA segment is looped out. It will be interesting to assess whether dimerization of repressors and the ensuing loop formation can enhance repression in eukaryotes too.

It is not clear which TFs have the propensity for the long-range effects and understanding the underlying rules will be important to reduce these indirect off-target effects. A recent bioinformatic analysis suggests that there are two classes of TFs. The TFs with short- and long-range regulatory influence differ in their chromatin-binding preferences and auto-regulatory properties [106]. The regulatory range is further affected by the 3D structure of the chromatin. Since many factors influencing transcription, such as activator and repressor domains and chromatin modifications can form aggregation bodies or other types of entities in which phase separation may play a role [91,107], the models constructed to explain short- and long-range effects may have to take these phenomena into consideration.

## Figures and Tables

**Figure 1 molecules-25-01902-f001:**
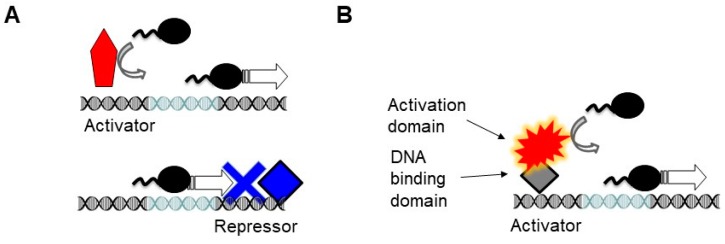
Modularity in the eukaryotic transcriptional regulation. The light blue segment of the DNA denotes the sequence recognized by the RNA polymerase (e.g., TATA box). (**A**) In prokaryotes, the RNA polymerase binds to the core promoter directly, which can be further enhanced by activators (red), which usually lack a separate activation domain. Prokaryotic repressors (blue) are simply DNA binding proteins, which act by blocking the binding of the polymerase along the DNA or its progression as a roadblock. Thus, they must bind at or downstream of the RNA polymerase binding sites. While the activator helps recruit the polymerase, the function of the repressor is solely to act as a DNA binding protein. (**B**) In eukaryotes, most activators have two domains. The DNA binding domains tether the activation domain to the DNA. The spiky shape denotes the tendency of the activation domains to have a disordered structure. Even a prokaryotic repressor can act as a DNA binding domain (gray diamond). Such modularly built activators are largely unknown in prokaryotes.

**Figure 2 molecules-25-01902-f002:**
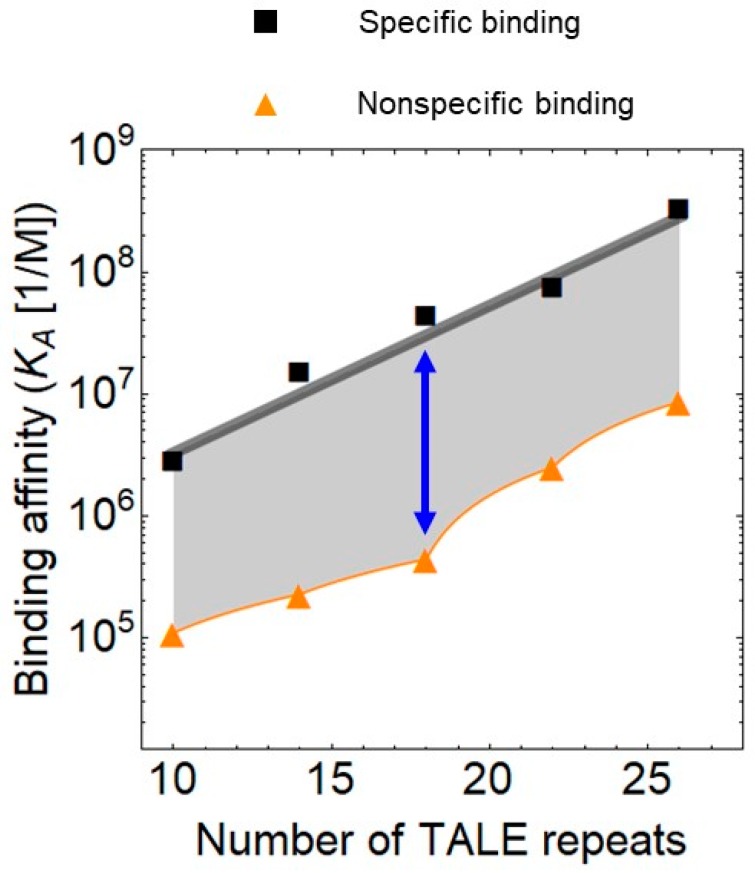
Specific and non-specific binding of transcription activator-like effectors (TALEs) to DNA as a function of the number of repeats, plotted using data from Rinaldi et al. [47]. The blue arrow denotes the 18 repeat long TALE array, which has the lowest relative non-specific binding to DNA.

**Table 1 molecules-25-01902-t001:** Therapies or clinical trials employing control of gene expression.

**Disease**	**Therapeutic Setting**
**Lipoprotein lipase (LPL) Deficiency**	
A rare autosomal recessive lipid disorder (1:1000,000). The failure to produce active LPL protein causes severe hypertriglyceridemia, associated with a high incidence of life-threatening acute pancreatitis attacks. In female patients, the disease is manifested during pregnancy.	Alipogene tiparvovec (Glybera), the first human gene therapy administered, results in sustained expression of the human LPL gene in muscle cells. The adenoassociated viruses that carry the LPL gene were injected intramuscularly. The number of pancreatitis attacks was reduced (but not eliminated) after the gene therapy [26,31,32].
**Peripheral Diabetic Neuropathy**	
	(Clinical Trial)
A common complication of diabetes. The gradual decline of the functionality of the microvasculature leads to poorer neuronal signal conduction in the affected extremities, causing pain and/or loss of sensation. Consequently, diabetic neuropathy sufferers are vulnerable to serious injury and infection.	To promote the formation of new blood vessels (revascularization), a plasmid encoding three zinc finger proteins that target a site in the vascular endothelial growth factor A (VEGFA) gene was injected intramuscularly. The zinc fingers were linked to a p65 transcriptional activator. The therapy proved safe, with only minimal adverse effects but with small, non-significant, benefit relative to the placebo group [27,33,34].
**Relapsed Leukemia**	
By the simultaneous introduction of the CAR and disruption of TCR and CD52 in T cells, functional CAR T cells were generated that could evade host immunity in the unmatched recipients. Such a combination is important for patients who do not have sufficient healthy T cells, which can occur in cases of relapsed leukemia.	Lentiviruses transduced the gene encoding CAR19 into the cells, which were then subjected to electroporation of TALEN mRNA targeting TRAC and CD52. Thereafter, residual TCR-expressing cells were depleted [35].

**Table 2 molecules-25-01902-t002:** Application of endogenous transcription factors (TFs).

**Cell Type/TF**	**Outcome**
**Control of TFs by Synthetic Gene Expression Systems**	
Expression of the proneuronal TF ASCL1 in glioblastoma stem cells under the control of the tetON promoter (stable transduction/piggyBac transposon) [39].	Activates neurogenic gene expression program and induces terminal differentiation, which may help the therapy of glioblastoma.
**Control of TFs by Small Molecules**	
The addition of the flavonoid Oroxylin A induces the expression of the TF HNF-4α (hepatocyte nuclear factor 4 alpha) [40].	The expression of HNF-4α target genes leads to the differentiation of a model hepatome, blocking cancer progression.

**Table 3 molecules-25-01902-t003:** TALE binding.

**Method of Measurements**	**Results**
**Relation Between Binding Affinity and Transcriptional Activation [68]**	
The binding affinities were measured in vitro with Electrophoretic mobility shift assay (EMSA), while the transcriptional activation was measured with TALE-VP64 fusions.	The apparent *K_D_* spanned four orders of magnitude, from 0.16 nM to 1800 nM.
**In Vivo Residence Time of TALEs with Varying Numbers of Repeats [70]**	
The DNA residence time of the TF was quantified in vivo in U2-OS cells by single molecule imaging of the individual TFs labeled with an organic dye.	The residence times of TALEs comprising 5, 7, 9, 13, 16 and 21 repeats ranged from 3 to 16 s. The 21-repeat TALE had intermediate residence time while the shortest TALE (5 repeats) had the longest residence time.
**The Effect of Increasing Numbers of TALE Repeats on the DNA Binding Specificity [47]**	
The binding affinities were measured with Electrophoretic mobility shift assay (EMSA) in the presence of magnesium and with fluorescence anisotropy (FA) in the absence of magnesium (150 mM NaCl).	Target specific binding is around 30 times stronger than binding to random sequences (in the presence of magnesium). In the absence of magnesium, the nonspecific binding is ten times stronger.
**In Vitro Binding Kinetics of TALEs [69]**	
FRET was used to study the in vitro binding of TALEs to DNA, with each of them being labelled with fluorescent dyes.	The bimolecular microscopic binding rate constant is 0.4 nM^−1^s^−1^ and the microscopic unbinding rate constant 0.3 s^−1^ for a 16-repeat TALE.

**Table 4 molecules-25-01902-t004:** CRISPR/Cas9 binding.

**Method of Measurements**	**Results**
**Enzymatic and Biophysical Characterization of DNA Cleavage by CRISPR/Cas9 [75]**	
The binding affinities were measured with double tethered DNA curtains and the binding events of quantum dot labelled Cas9-guide RNA were recorded with total internal reflection fluorescence microscopy.	*K_D_* = 0.5 nM for the Cas9-guide RNA binding to the target DNA. Even without guide RNA, the upper limit of binding is 25nM for the apo-Cas9–DNA complex.
**Detailed Kinetic Characterization of the CRIPSR/Cas9 Binding to the DNA [77]**	
Quench flow experiments were performed by mixing the Cas9-guide RNA complex with 32P-labeled DNA substrate. The reaction was stopped by the addition of EDTA at varying time points. The products were separated by polyacrylamide gel electrophoresis.	The dissociation rate constant of DNA from Cas9.gRNA.DNA (k_off_ = 0.0024 s^−1^) is equivalent to a residence time of around 5 min. For the equilibrium DNA binding, *K_D_* = 4 nM.
**In Vivo Residence Time as a Function of Varying Guide RNA Lengths [78]**	
Directly labeled guide RNA (with Broccoli aptamer) and mCherry labelled dCas9 were used to track the binding at their target-site using fluorescence recovery after photobleaching (FRAP) measurement. The guide RNA was targeted to a unique sequence at the subtelomeric region.	The residence time and the off-rate of the dCas9/C3-11–guide RNA complex on the C3 target were estimated to be 206 min and 2.9 × 10^−4^ s^−1^. When the guide RNA length was truncated from 11 to 8 nucleotides, the residence time decreased from 206 to 25 min.
**In Vivo Binding Kinetics of Cas9d in *E. coli* [79]**	
dCas9 fused to a fluorescent protein was expressed at a low copy numbers (about five molecules per cell). The DNA-bound fluorophores were detected as diffraction-limited spots with single-molecule fluorescence microscopy.	The association rate is 2.7 × 10^−3^ min^−1^ molecule^−1^ while the dissociation time varied between 40 and 120 min, depending on the growth condition.

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
