# Peer review of "Tuning up Transcription Factors for Therapy"

_molecules, 2020, doi:10.3390/molecules25081902_

Round 1

Reviewer 1 Report

Manuscript Number: Molecules-745511

Title: Tuning up transcription factors for therapy

In this review manuscript the authors complied information regarding the development of transcription factors and their therapeutic applications. The review is well organized in categories of the modularity principle in sequence recognition. Overall the manuscript is appealing, original, and should be accepted after revision.

Specific comments:

  • Section 2.1.4: First line indent is missing.
  • Figure 1: The illustration should be improved to facilitate comprehension.
  • Figure 1 legend: It is missing the color code description. Please, remove double punctuation in the end of paragraph.
  • Table 1: Remove double punctuation in the end of paragraph.
  • Section 5: First line indent is missing.
  • Box 1: Legend is incomplete.
  • Box 1 should be removed.
  • Section 7 should be consolidated in section 2 to improve readability.
  • Add space in the end of table 3 and following text.
  • Figure 2 should be removed.

Author Response

In this review manuscript the authors complied information regarding the development of transcription factors and their therapeutic applications. The review is well organized in categories of the modularity principle in sequence recognition. Overall the manuscript is appealing, original, and should be accepted after revision.

I thank the reviewer for the comments, which helped to improve the manuscript.

Specific comments:

  • Section 2.1.4: First line indent is missing.

Done.

  • Figure 1: The illustration should be improved to facilitate comprehension.

The eukaryotic repressor is removed and the focus is on the modularity in eukaryotes.

  • Figure 1 legend: It is missing the color code description. Please, remove double punctuation in the end of paragraph.

The colors are now described in the legend.

  • Table 1: Remove double punctuation in the end of paragraph.

Done.

  • Section 5: First line indent is missing.

Corrected.

  • Box 1: Legend is incomplete.
  • Box 1 should be removed.

Box and legend are removed.

  • Section 7 should be consolidated in section 2 to improve readability.

The Section 7.2 is rewritten as follows:

“Three molecules interact to yield the Cas9-CRISPR RNA-DNA complex, and the binding to the DNA is followed by an enzymatic reaction. Thus, a more complex kinetics is expected for this reaction than for the bimolecular binding reaction between the TALE and DNA. The first kinetic analysis of the Cas9-CRISPR complex revealed an even more complicated kinetics: the DNA cleavage failed to obey the Michaelis-Menten kinetics [74]. The cleavage reaction stopped soon after mixing the components, leaving a large proportion of DNA uncut, even though a usual enzymatic reaction proceeds until completion. A higher proportion of DNA was cleaved only if the Cas9-RNA concentration was increased”

  • Add space in the end of table 3 and following text.

Done

  • Figure 2 should be removed. 

The shading has been changed so that the difference between the specific and nonspecific binding becomes more clear.

Reviewer 2 Report

This review focuses on describing the current toolbox commonly used for artificial transcriptional regulation. Overall, the manuscript provides a concise summary of the biochemical and biophysical properties of these designer regulators that can be explored to modulate the efficacy and specificity. The review can be enhanced by addressing the following points:

  1. Since not all systems described in the review (such as the CRISPR-Cas system) are transcriptional factors in nature, the title is a bit of misleading and should be revised.
  2. This review cites quite a few other reviewer papers. While this is difficult to completely avoid, if possible, the original studies, especially some of the landmark publications should be cited.
  3. When discussing the difference between prokaryotic and eukaryotic repressors, it may be useful to contrast briefly the chromosome structures between the two.
  4. The schematic of transcriptional regulation shown in Figure 1 is too simplistic, especially 1B for eukaryotes, even by text-book standards.
  5. The manuscript will also be benefitted from professional English editing.
  6. There are quite a few misspelling and grammatical errors in the text.
  7. Line 61: “endonuceases” should be “endonucleases”
  8. Lines 203 and 204: “are can” should be “can”?
  9. Box 1: the binding equilibrium equation for the P and DNA interaction with kinetic constants should be provided at the beginning for better clarity.
  10. Line 403: “increasing reducing” should be “reducing”?
  11. Line 420: “binding and unbinding rate constants” maybe “on and off-rate constants”
  12. Lines 457-460: this sentence is very confusing and should be rewritten.

Author Response

This review focuses on describing the current toolbox commonly used for artificial transcriptional regulation. Overall, the manuscript provides a concise summary of the biochemical and biophysical properties of these designer regulators that can be explored to modulate the efficacy and specificity. The review can be enhanced by addressing the following points:

  1. Since not all systems described in the review (such as the CRISPR-Cas system) are transcriptional factors in nature, the title is a bit of misleading and should be revised.

Now, the Cas was changed to dCas in the abstract, so that it becomes immediately clear that we deal with the catalytically dead version of Cas. This review covers both natural and synthetic (artificial) transcription factors (for example tetR-VP16 (tTA)), so the dCas9 is covered by the title.

  1. This review cites quite a few other reviewer papers. While this is difficult to completely avoid, if possible, the original studies, especially some of the landmark publications should be cited.

We tended to cite the original papers when they have been published recently. Now, we have added the reference for the dCas9 and the landmark TALE articles.

  1. When discussing the difference between prokaryotic and eukaryotic repressors, it may be useful to contrast briefly the chromosome structures between the two.

A paragraph is now added that describes the differences in the chromosomal structure:

“The prokaryotic DNA is naked while the eukaryotic DNA is wrapped around histones. In line with this difference, the mass ratio of the basic proteins to DNA is 50 times higher in eukaryotes than in prokaryotes [49]. The naked prokaryotic DNA is easily accessible to the transcriptional machinery while the default state of the eukaryotic DNA is restrictive due to lower accessibility of the DNA in the chromatin. Therefore, transcriptional activators are essential to recruit the RNA polymerase and to initiate transcription in eukaryotes, while they fine tune the gene expression in prokaryotes, playing a subordinate role in relation to repressors [50]. The eukaryotic TFs recruit enzymes that modify chromatin, including acetylation and methylation, and these epigenetic modifications influence the DNA accessibility. Thus, indirectly, repressors usually render the DNA less accessible while activators have the opposite effect.”

  1. The schematic of transcriptional regulation shown in Figure 1 is too simplistic, especially 1B for eukaryotes, even by text-book standards.

The title of the figure has been changed to “Modularity in the eukaryotic transcriptional regulation” in order to indicate that the sole purpose of this scheme is to indicate the absence and presence of modularity in transcriptional regulation, and not to compare transcriptional regulation in general. Similarly, now we indicate that the spiky shape denotes the intrinsically disordered structure.   

  1. The manuscript will also be benefitted from professional English editing.
  2. There are quite a few misspelling and grammatical errors in the text.

The text have been thoroughly revised to remove convoluted sentences, grammatical errors and typos.

  1. Line 61: “endonuceases” should be “endonucleases”

This typo is now corrected.

  1. Lines 203 and 204: “are can” should be “can”?

This mistake is now corrected.

  1. Box 1: the binding equilibrium equation for the P and DNA interaction with kinetic constants should be provided at the beginning for better clarity.

The equation denoting the dimerization reaction is now included.

  1. Line 403: “increasing reducing” should be “reducing”?

The fragmented sentence is now corrected.

  1. Line 420: “binding and unbinding rate constants” maybe “on and off-rate constants”

Both versions are accepted, as indicated in the section explaining the kinetic equations.

“The association (binding) and dissociation (unbinding) rate constants of this binding reaction are denoted by …”

Both versions are mentioned in this review since the cited articles use one or the other term.

  1. Lines 457-460: this sentence is very confusing and should be rewritten.

This sentence has been rewritten and put in the correct context:

“Thus, as cells differentiate from stem cells to neurons, the epigenetic activity gradient around the recruited TALE-VP160 broadens. The VP16 (or its variants VP64, VP160) domain, which contains a large number of negatively charged amino acids, may cause these epigenetic changes independently of transcription, since the recruitment of negatively charged peptides to the DNA can cause large-scale chromatin remodeling, without inducing transcription [101]”

Round 2

Reviewer 2 Report

The authors changed Cas to dCas in the abstract to emphasize the catalytically dead version of Cas and its utility in transcriptional regulation. However, strictly speaking, CRISPR-dCas9 is an artificial transcription regulator, not a transpiration factor. This reviewer still thinks that it is more appropriate to use a broader title such as: “Tuning up transcription regulators for therapy.”

Author Response

Yes, I think that the term transcription factor covers both natural and synthetic transcription factors. To make this even clearer, now I changed the last sentence in the abstract to "Understanding these mechanisms will help to tailor natural and synthetic transcription factors to the needs of specific applications.". The term "synthetic" is not mentioned in the title because the review covers also natural transcription factors albeit to a lesser extent.